# Exploring Protein-Inorganic Hybrid Nanoflowers and Immune Magnetic Nanobeads to Detect *Salmonella* Typhimurium

**DOI:** 10.3390/nano8121006

**Published:** 2018-12-04

**Authors:** Lei Wang, Xiaoting Huo, Ruya Guo, Qiang Zhang, Jianhan Lin

**Affiliations:** 1Key Laboratory of Agricultural Information Acquisition Technology, Ministry of Agriculture, China Agricultural University, Beijing 100083, China; wanglei123@cau.edu.cn (L.W.); huoxiaoting@cau.edu.cn (X.H.); 2Key Laboratory of Modern Precision Agriculture System Integration Research, Ministry of Education, China Agricultural University, Beijing 100083, China; guoya@cau.edu.cn; 3Department of Biosystems Engineering, University of Manitoba, Winnipeg, MB R3T 2N2, Canada; Qiang.Zhang@umanitoba.ca

**Keywords:** protein-inorganic hybrid nanoflower, continuous-flow potentiometric detection, ion selective electrode, *Salmonella*

## Abstract

Early screening of pathogenic bacteria is key to preventing and controlling outbreaks of foodborne diseases. In this study, protein-inorganic hybrid nanoflowers were synthesized for signal amplification and used with a calcium ion selective electrode (Ca-ISE) to establish a new enzyme-free assay for rapid and sensitive detection of *Salmonella*. Calcium hydrophosphate crystals were first conjugated with polyclonal antibodies against *Salmonella* to synthesize immune calcium nanoflowers (CaNFs), and streptavidin modified magnetic nanobeads (MNBs) were conjugated with biotinylated monoclonal antibodies against *Salmonella* to form immune MNBs. After target bacteria were separated using immune MNBs to form magnetic bacteria, immune CaNFs were conjugated with magnetic bacteria to form nanoflower conjugated bacteria. Then, hydrogen chloride was used to release calcium ions from nanoflower conjugated bacteria. After magnetic separation, the supernatant was finally injected as a continuous-flow to fluidic chip with Ca-ISE for specific detection of calcium ions. The supernatant’s potential had a good linear relationship with bacteria concentration, and this assay was able to detect the *S.* Typhimurium cells as low as 28 colony forming units/mL within two hours. The mean recovery of target bacteria in spiked chicken samples was 95.0%. This proposed assay shows the potential for rapid, sensitive, and on-line detection of foodborne pathogens.

## 1. Introduction

Foodborne pathogens have become a major public health issue, increasingly attracting concern around the world. The WHO reported that almost 10% of the world population falls ill annually due to the consumption of foods and water contaminated by pathogenic microorganisms [1]. *Salmonella* is one of the main foodborne pathogens, which passes through the entire food supply chains from livestock farming to food processing, then to food logistics, and finally to retail. The clinical manifestations of *Salmonella* poisoning mainly include acute onset of fever, abdominal pain, diarrhea, nausea, and sometimes vomiting. Therefore, early screening of *Salmonella* and other common foodborne pathogens is vital to ensure food safety in food supply chains.

Ion selective electrode (ISE) is a type of electrochemical sensor that converts the activity of specific ions in a solution into the potential to determine the ion concentration [2,3]. Various ISEs have been developed and widely used in environmental monitoring [4,5,6], biochemical analysis [7,8,9,10], food safety [11,12], and other applications [13,14] due to their unique advantages, such as low cost, simple operation, small size, rapid response, and good selectivity. In addition, ISEs do not require external electrical excitement during electrochemical measurements; thus they are the most straightforward tool compared to other electrochemical methods [15,16]. ISEs are generally used to measure the specific ions in a static solution, but seldom under continuous-flow conditions. Recently, dynamic potentiometry, which was first introduced by Calvo et al. [17], has gained popularity. Some studies on the dynamic behaviors of ISEs were reported in developing new kinetic potentiometric methods [18]. Cuartero et al. reported a dynamical potentiometric ISE using 18-crown-6 as ionophore for continuous-flow detection of multiple ions in binary mixtures [19]. Thus, combining ISE-based potentiometric methods with the immune methods is showing promise for the development of novel assays using some specific ions as a signal. 

In recent years, fast development of various nanomaterials has opened up new possibilities for biosensing signal amplification. Among them, protein-inorganic hybrid nanoflowers have been demonstrated to be able to maintain or even increase the activity of the proteins and effectively amplify the detection signals [20,21,22]. Ge et al. first proposed the coprecipitation method to synthesize hybrid organic-inorganic nanoflowers using copper ions as an inorganic component and proteins as an organic component. They successfully used laccase as the model protein to create the nanoflowers for detection of epinephrine [23]. Ye et al. developed Con A-invertase-CaHPO_4_ hybrid nanoflowers to enhance the activity of the immobilized invertase and used them with a personal glucose meter for sensitive detection of *Escherichia coli* O157:H7 [24]. Ismail et al. reported the synthesis of hybrid nanoflowers using iron ions (Fe^2+^) and horseradish peroxidase (HRP) and demonstrated that they increased the catalytic activity of HRP more than five-fold [25]. Therefore, the introduction of nanoflowers might further improve the sensitivity of the potentiometric assays.

In this study, we explore a new enzyme-free assay to detect *S.* Typhimurium using protein-inorganic hybrid nanoflowers for effective amplification of biological signals, magnetic nanobeads for immunomagnetic separation of target bacteria, and fluidic chip with the calcium ion selective electrode for continuous-flow detection of calcium ions. The objectives of this study were to: (1) develop a new enzyme-free, potentiometric assay for rapid and sensitive detection of *S.* Typhimurium, and (2) experimentally evaluate the sensitivity and applicability of the assay. As shown in Scheme 1, prior to testing, the streptavidin-modified magnetic nanobeads were conjugated with the biotinylated monoclonal antibodies against *Salmonella* to form the immune magnetic nanobeads through streptavidin-biotin binding, and calcium hydrophosphate crystals were conjugated with the polyclonal antibodies against *Salmonella* to synthesize the immune calcium nanoflowers through coprecipitation method. First, the immune magnetic nanobeads were used to separate the target bacteria from the sample to form the magnetic nanobeads-bacteria complexes (magnetic bacteria) that were concentrated in a smaller volume of phosphate buffered saline (PBS). Then, the immune calcium nanoflowers were conjugated with the magnetic bacteria to form the magnetic nanobeads-bacteria-nanoflower complexes (nanoflower conjugated bacteria). Successively, hydrogen chloride was used to release calcium ions from nanoflower conjugated bacteria. After magnetic separation, the supernatant was finally injected to the fluidic chip for continuous-flow detection of calcium ions to determine the concentration of the target bacteria.

## 2. Materials and Methods

### 2.1. Materials

The anti-*Salmonella* monoclonal antibodies (concentration: 1 mg/mL) from Abcam (Cambridge, MA, USA) and the anti-*Salmonella* polyclonal antibodies (concentration: 2.5 mg/mL) from Meridian (Memphis, TN, USA) were used for specific conjugation with the target *S.* Typhimurium cells. The long-arm biotin labeling kit from Elabscience Biotechnology (Wuhan, China) was used for the modification of biotin onto the monoclonal antibodies (MAbs). The streptavidin-modified magnetic nanobeads from Ocean Nanotech (SV0152, Fe content: 1 mg/mL, San Diego, CA, USA) were used with the biotinylated MAbs for immunomagnetic separation of the target bacteria. Calcium chloride from XiLong Scientific (Shantou, China) was used for the synthesis of the protein-inorganic hybrid nanoflowers. Phosphate buffered saline from Sigma Aldrich (10 times concentrated, St. Louis, MO, USA) was diluted with the deionized water to prepare the PBS solution. Bovine serum albumin (BSA) also from Sigma Aldrich (St. Louis, MO, USA) was used for blocking (1% and 10%, *w*/*v*). Tween 20 from Amresco (Solon, OH, USA) was used for washing. A silicone elastomer kit from Dow Corning (Sylgard 184, Auburn, MI, USA) was used for fabricating the poly(dimethoxy)silane (PDMS) channels. The printing material (Vero Whiteplus RGD835, Stratasys, Eden Prairie, MN, USA) was used with the Objet24 three-dimensional (3D) printer for fabricating the mold of the PDMS channels. The deionized water was produced by Millipore Advantage 10 (18.2 MΩ∙cm, Billerica, MA, USA) and used to prepare all the solutions.

### 2.2. Synthesis of Protein-Inorganic Nanoflowers for Signal Amplification

The synthesis of protein-inorganic hybrid nanoflowers was based on the previously reported coprecipitation method with modifications [23,26]. First, 0.02 mg polyclonal antibodies (PAbs) against *Salmonella* and 0.18 mg BSA were simultaneously added into 1 mL PBS (3 mM). Then, 20 μL calcium chloride (200 mM) was added. After incubation at 15 rpm for 12 h, the immune calcium nanoflowers (CaNFs) were formed. To remove the surplus calcium ions and PAbs, the immune CaNFs were centrifuged at 15,000 rpm for 5 min, resuspended in the PBS (3 mM), and shaken evenly for 1 min to wash the immune CaNFs. The washing step was repeated 3 times to ensure that the free calcium ions and antibodies were washed away thoroughly. Finally, the immune CaNFs were resuspended in PBS and stored at 4 °C for further use.

### 2.3. Fabrication of Fluidic Chip for Potentiometric Detection

Fluidic chip with a calcium ion selective electrode (Ca-ISE) was developed for potentiometric detection of calcium ions in a continuous-flow condition. As shown in Appendix A, the fluidic chip consisted of Ca-ISE (9720BNWP, Thermo Fisher, Waltham, MA, USA) for potentiometric detection of calcium ions; a fluidic channel 30 mm long, 2 mm wide, and 2 mm tall for continuous-flow transportation of the supernatant; and a detection chamber with a 4 mm diameter for tight housing of the Ca-ISE. The 3D structural mold of the fluidic channel and the detection chamber were first designed using Solidworks (Dassault Systèmes Solidworks Corp., Waltham, MA, USA) and fabricated using the Objet24 3D printer. Then, the silicone elastomer base and curing agent were uniformly mixed at the ratio of 10:1 to cast the PDMS channel, which was bonded with the glass slide to fabricate the fluidic chip. Finally, the Ca-ISE was tightly inserted into the detection chamber, and the membrane of the Ca-ISE remained at the same level with the top of the channel for sensitive measurement and efficient washing.

### 2.4. Preparation of the Bacterial Cultures

*Salmonella* Typhimurium (ATCC14028) was used as the target bacteria, and *E. coli* O157:H7 (ATCC43888) and *Listeria monocytogenes* (ATCC13932) were used as the non-target bacteria. They were cultured in the Luria-Bertani (LB) medium (Aoboxing Biotech, Beijing, China) overnight at 37 °C with shaking at 180 rpm. The bacteria were 10-fold diluted with sterile PBS to obtain concentrations from 10^2^ to 10^6^ colony forming units (CFU)/mL. For bacteria enumeration, the bacteria were serially diluted with the sterile PBS and were grown on the LB plates. After incubation at 37 °C for 24 h, the visible colonies were counted to determine the concentration of the bacteria.

### 2.5. Bacteria Separation and Detection

Prior to testing, the anti-*Salmonella* monoclonal antibodies (MAbs) were modified with biotin through membrane dialysis using the long-arm biotin labeling kit according to the manufacturer’s protocol to obtain the biotinylated MAbs. For magnetic separation and concentration of the target bacteria, 20 μL of the streptavidin modified magnetic nanobeads (MNBs) were first washed with 500 μL of PBST (PBS containing 0.05% Tween 20). Transmission electron microscopy (TEM) imaging was conducted to characterize the MNBs (Figure 1a). Then, 4 μL of the biotinylated MAbs were added to conjugate with the MNBs at 15 rpm for 45 min. After washing with PBST to remove the excessive antibodies, the immune MNBs were obtained. Finally, different concentrations of 500 μL of the target bacteria were added to resuspend the immune MNBs, and then incubated at 15 rpm for 45 min to form the magnetic bacteria samples of different concentrations. TEM imaging was conducted to characterize the magnetic bacteria (Figure 1b).

For detection of the magnetic bacteria, 100 μL of the immune CaNFs were added and incubated with the magnetic bacteria for 45 min, allowing the formation of the nanoflower conjugated bacteria. After the nanoflower conjugated bacteria were transferred to a new tube and washed with PBST 3 times, then 500 μL hydrogen chloride was added into the nanoflower conjugated bacteria, incubated at 15 rpm for 5 min and magnetically separated for 3 min to obtain the supernatant containing calcium ions through substitutional reaction. Finally, the supernatant was continuously injected using the syringe pump (Pump 11 Elite, Harvard Apparatus, Holliston, MA, USA) into the fluidic chip at a flow rate of 250 μL/min. The potential of the Ca-ISE was measured on-line using a potentiometer (M555P, Pinnacle, Corning, NY, USA) and recorded every 15 s.

### 2.6. Bacteria Detection in Chicken Carcass

Prior to testing, each chicken carcass purchased from a local supermarket was placed in a sterile plastic bag containing 250 mL of PBS (pH 7.4, 10 mM), followed by vigorous shaking for 1 min to rinse the chicken carcass and standing for 10 min to obtain the supernatant. First, 1 mL of different concentrations of the target bacteria were added into 9 mL supernatant to prepare the spiked chicken samples with the bacterial concentrations from 10^2^ to 10^6^ CFU/mL. Then, the immune MNBs were used to magnetically separate the target bacteria from the spiked samples with different concentrations, respectively. After the magnetic bacteria were washed with PBST to avoid non-specific reaction, the immune CaNFs were added and incubated to form the nanoflower conjugated bacteria, and hydrogen chloride was used to resuspend the nanoflower conjugated bacteria to release calcium ions. Finally, the supernatant was obtained by magnetic separation and injected into the fluidic chip for potential measurement in a continuous-flow condition to determine the concentration of the target bacteria.

## 3. Results and Discussion

### 3.1. Characterization of Calcium Nanoflowers

The synthesis of the calcium nanoflowers based on the facile one-step coprecipitation method is the key to the development of the proposed assay. The primary crystals of calcium hydrophosphate were first formed through chemical reaction between calcium ions and hydrophosphate ions. After the proteins (BSA and PAbs) were added, the petals containing calcium ions were then formed through the coordination of amide groups on the protein backbone [23]. Finally, the immune calcium nanoflowers were completely formed by anisotropic growth, i.e., the proteins induced the nucleation of the calcium hydrophosphate (CHP) crystals to form the scaffold of the nanoflowers and acted as the cement to conjugate the petals. As observed from the TEM images (Figure 1c), the synthesized CaNFs had a diameter of ~1.50 µm. The dynamic light scatting technique was used to further characterize the diameter of the nanoflowers, and the result shown in Figure 1d indicates that the diameter of the nanoflowers was ~1.42 µm, which was consistent with the result of TEM imaging.

To further investigate the amount of calcium ions in the CaNFs, different concentrations of CaCl_2_ ranging from 10 µM to 500 µM were measured using the flame photometer (Model 410, Sherwood Scientific, Cambridge, UK) and a linear calibration curve was developed (Appendix A). The calcium concentration of the synthesized CaNFs was measured using the flame photometer and calculated to be ~1.41 mM. 

### 3.2. Evaluation on Ca-ISE for Continuous-Flow Detection of Calcium Ions

In our previous studies, interdigitated array microelectrodes [27,28,29], a screen-printed interdigitated electrode [30], and a printed circuit board electrode [31] were used as transducers for the development of electrochemical impedance biosensors. However, these electrodes could not distinguish the specific ions and often suffered from serious interference from both the sample background and the buffer solution, leading to low signal-to-noise ratios and sensitivity. The introduction of ISE with better selectivity to specific ions might improve the robustness and sensitivity of our previous electrochemical biosensors. Thus, the Ca-ISE was employed for specific detection of the signal (calcium ions) in this study. 

Since this study aimed to develop a potentiometric assay for on-line detection of *S.* Typhimurium using calcium ions as a signal, it was essential to verify the feasibility of Ca-ISE for continuous-flow detection of calcium ions, which has traditionally been used in static conditions. Deionized water was first injected into the fluidic chip at the flow rate of 250 μL/min for cleaning and the measurements were used as the control. Then, different concentrations (10–500 μM) of CaCl_2_ were injected into the chip from low concentration to high concentration at the same flow rate and the potentials were recorded every 15 s after continuous-flow injection for one minute. After each concentration of CaCl_2_ was measured, the chip was thoroughly washed with deionized water for three minutes until the potential returned to the original level (the potential of the deionized water). As shown in Figure 2a, when the concentration of CaCl_2_ changed from 10 μM to 500 μM, the potential measured in a continuous-flow condition increased from −73.7 mV to −28.2 mV. The potential remained fairly stable when the CaCl_2_ solution was continuously injected, and returned to the original level after continuous-flow washing. As shown in Figure 2b, the average potential (*E*) of the Ca-ISE was found to have a good linear relationship with the concentration (*C*) of calcium ions, which could be expressed by *E* = 11.89ln(*C*) − 102.98 (*R*^2^ = 0.98). The successful implementation of continuous-flow detection of calcium ions enabled the integration of the Ca-ISE to develop a lab-on-a-chip assay for on-line monitoring of foodborne bacteria.

### 3.3. Optimization of Proposed Assay

The release of calcium ions from different concentrations of the nanoflower conjugated bacteria was the basis of this proposed assay. Therefore, it was necessary to investigate the impact of the washing time of the CaNFs on the background noise control. The immune CaNFs were centrifuged at 15,000 rpm for 5 min to obtain the washing solution, and resuspended in 1 mL PBS (3 mM). As shown in Figure 3a, the potential of the washing solution decreased from −16.60 mV to −66.97 mV after three washes and changed little after the third wash, indicating that most non-specific absorbed calcium ions were washed away after washing three times. To minimize the impact of the noise from the background, the CaNFs were washed three times prior to use in this study. 

The amount of the CaNFs and the concentration of hydrogen chloride considerably impact the release of calcium ions from the nanoflower conjugated bacteria and are important to the sensitivity of the proposed assay. Different concentrations of the CaNFs (Ca^2+^ concentration: 0.35–1.41 mM, 100 μL) were used to conjugate with the *S.* Typhimurium cells at a concentration of 7.2 × 10^5^ CFU/mL. The potential of the supernatant was measured in a continuous-flow condition using the fluidic chip. As shown in Figure 3b, when the concentration of CaNFs increased from 0.35 mM to 1.41 mM, the potential decreased from −50.83 mV to −41.57 mV, indicating that more CaNFs were conjugated with the target bacteria at higher concentrations of CaNFs. However, further increase in the concentration of the CaNFs resulted in little change (<3%) in the potential. Therefore, the optimal concentration of 1.41 mM for the CaNFs was used in this study. 

Calcium ions on the nanoflower conjugated bacteria have to be released from the bacteria since the Ca-ISE only responds to free calcium ions. Thus, HCl was used in this study to replace calcium ions on the nanoflower conjugated bacteria with hydrogen ions. Different concentrations of HCl (50 μM–10 mM, 500 µL) were reacted with 100 µL CaNFs and the potential was dynamically measured by the fluidic chip. As shown in Figure 3c, the potential increased from −61.97 mV to −39.03 mV since more free calcium ions were replaced, while the concentration of HCl increased from 50 μM to 1 mM. However, further increase in the HCl concentration from 1 to 10 mM only resulted in a slight (4%) increase in the potential. Therefore, the optimal concentration of 1 mM for HCl was used in this study.

### 3.4. Detection of S. Typhimurium in Pure Cultures and Spiked Chicken Samples

To determine the unknown concentration of *S.* Typhimurium in a sample using this proposed assay, the calibration model between the potential and the concentration was established. Thus, three parallel tests on pure *S.* Typhimurium cells with different concentrations of 10^2^–10^6^ CFU/mL were conducted using this proposed assay. As shown in Figure 4a, a linear relationship between the potential (*E*) of the Ca-ISE in a continuous-flow condition and the concentration (*C*) of the *S.* Typhimurium cells was found, which could be expressed as *E* = 2.33ln(*C*) − 73.07 (*R*^2^ = 0.98). The low detection limit of this assay was determined to be 2.8 × 10^1^ CFU/mL, according to three times of signal-to-noise ratio. The high sensitivity of this proposed assay could be attributed to three aspects: (1) effective amplification of the biological signals using the protein-inorganic hybrid nanoflowers with larger surface-to-volume ratio, resulting in more calcium ions on the nanoflower conjugated bacteria; (2) specific detection of calcium ions using the Ca-ISE, resulting in better control of the background noise; and (3) effective washing of the Ca-ISE due to continuous-flow and high-pressure flushing on the ion selective membrane, resulting in less interference from both the buffer solution and the sample background. TEM imaging confirmed the successful formation of the nanoflower conjugated bacteria, as shown in Figure 4b.

To further evaluate the practical applicability of the proposed assay, three parallel tests on different concentrations of the *S.* Typhimurium cells in the spiked chicken samples were conducted. As shown in Figure 4c, the potential of Ca-ISE for the spiked chicken samples containing the *S.* Typhimurium cells at different concentrations from 1.0 × 10^2^ to 1.0 × 10^6^ CFU/mL were slightly less than for the pure cultures at the same concentrations. This might be due to the interference from the background of the chicken samples, since a large number of proteins, fats, and other molecules in chicken might influence the separation of the target bacteria and the detection of calcium ions. The recovery of the target bacteria (*R*) was calculated as the ratio of the concentration of the target bacteria in the spiked chicken sample (*Ns*) to that of the bacteria in the pure culture (*Nc*), i.e., *R* = *Ns*/*Nc* × 100%. As shown in Table 1, the recoveries for different concentrations of the target bacteria ranged from 92.0% to 100.8%, with an average of 95.0%. This data clearly show that the proposed assay is adequate for detection of *S.* Typhimurium in the chicken samples. 

### 3.5. Specificity of Proposed Assay

In this study, *L. monocytogenes* and *E. coli* O157:H7 were used as non-target bacteria to evaluate the specificity of the proposed assay. The target bacteria (*Salmonella* Typhimurium) and these two non-target bacteria at the same concentration of 1.0 × 10^4^ CFU/mL and the negative controls (PBS) were detected using the proposed assay. As shown in Figure 5, the potentials of Ca-ISE for the negative controls, *L. monocytogenes,* and *E. coli* O157:H7 were −66.23 mV, −60.43 mV, and −58.58 mV, respectively, which were equivalent to the target bacteria with the respective concentrations of 1.9 × 10^1^, 2.3 × 10^2^, and 5.0 × 10^2^ CFU/mL, respectively, based on the calibration model. The potential for *S.* Typhimurium was −49.20 mV, which was equivalent to the target bacteria with a concentration of 2.8 × 10^4^ CFU/mL. The slight potential difference between *Listeria monocytogenes*, *E. coli* O157:H7, and the negative controls might be attributed to: (1) negligible cross reaction between the MAbs/PAbs and the *Listeria monocytogenes* and *E. coli* O157:H7 cells, resulting in a small amount of calcium ions; and (2) electrostatic adsorption of the CaNFs onto the non-target bacteria or impurities non-specifically captured by the MNBs, resulting in elevated background noises. The potential for the *S.* Typhimurium cells at the same concentration was much higher (more than two times compared to the control) than those for *Listeria monocytogenes* and *E. coli* O157:H7 due to the formation of the nanoflower conjugated bacteria resulting in the replacement of the nanoflowers to produce more calcium ions. This indicates that the proposed assay has good specificity. 

## 4. Conclusions

A novel potentiometric assay based on effective amplification of biological signals using protein-inorganic hybrid CaNFs, efficient separation of target bacteria using immune MNBs, and continuous-flow detection of calcium ions using Ca-ISE was successfully developed for rapid and sensitive detection of *Salmonella*. The proposed assay was able to detect *S.* Typhimurium cells as low as 28 CFU/mL within two hours. The Ca-ISE embedded in the fluidic chip is capable of online monitoring calcium ions and has potential for the development of a lab-on-a-chip potentiometric biosensing device for sensitive detection of foodborne pathogens.

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
