# Peer review of "Exploring Protein-Inorganic Hybrid Nanoflowers and Immune Magnetic Nanobeads to Detect Salmonella Typhimurium"

_nanomaterials, 2018, doi:10.3390/nano8121006_

Round 1
Reviewer 1 Report
This publication makes use of calcium hydrophosphate nanoflowers conjugated with antibodies against one particular bacterial species, which are further conjugated with magnetic nanobeads, resulting in nanoflower bacteria. The authors describe the use of a very sensitive calcium ion detection electrode for correlating the amount of bacteria captured based on the specific change in calcium ion potential recorded by an electrode. The authors derive their motivation based on previous publications, where specificity of ions detected by the electrode are concluded to be extremely sensitive in detecting the ion concentrations and hence avoiding background interference from the sample and the buffer. The authors show extremely well, based on the calculations, that they are able to capture 95% of food-borne bacteria, Salmonella, in pure as well as spiked chicken samples. Low concentrations of bacteria such as 28 CFU/ml can be captured and detected using this method. Changes in the potential of the chip recorded at different concentrations of Salmonella and HCl for calcium ion release, varied washing times of the electrode and different nanoflower concentrations were performed and summarized to postulate the experimental conditions. The specificity of the method was also confirmed using other bacterial species such as E.coli and Listeria. All in all, this is a well-explained paper based on the idea of developing simple lab-on-chip system for quick and easy detection of a particular bacterial species.
Further Comments:
Section 2.5: Methods section: Tween is usually used to prevent non-specific protein protein interactions for example, in western blot. The authors mention removal of excess of antibodies. Tween 20 is a non-ionic surfactant, which might have an harmful effect on cell membranes. At the same time, concentrated hydrochloric acid solution was used to release calcium ions from the nanoflower.
•Does Tween 20 and/or HCl in this experimental setup affect the viability of bacteria?
•Do HCl or Tween 20 cause the release of bacterial intercellular material due to the affect on their bacterial membrane? Is there a possibility that the damages in the bacterial membrane cause calcium to be released from cells hence affecting the potential recorded by the chip? The amount of calcium recorded in prokaryotic cells is about 100-300 nM as mentioned by Dominguez[1] which might affect the potential recorded by the calcium electrode
Membrane viability of the cells, before and after treatment should be confirmed.
Figure 1: Volume distribution of NFs: How do the values for number and intensity measurements at the dynamic light scattering are comparable? Based on the TEM size values, how many of the nanoflowers were observed to confirm the size of the nanoflower?
What is the zeta potential of the CaNF? This can be performed using DLS measurements. The charge on the particles might lead to binding of other components of chicken carcasses or the “non-target bacteria” used in this publication. Are there some tests, such as optical density measurements to confirm the specific attachment of CaNF to Salmonella only and not to other negatively charged bacterial surfaces and/or ions in the test solution.
Other:
Line 23: Is the terminology nanoflower bacteria appropriate? Or is it nanoflower conjugated bacteria or vise versa.
Line 113: resuspended “in” PBS
Line 116: “in a” continuous flow-condition
Missing:
Figure 1: better picture required! Comparable with number or intensity value graphs?
Picture of magnetic nanobeads without/with bacteria-missing!
Method for observation of bacteria using TEM missing-Methods section-missing!
Figure 2(b)- Salmonella along with nanoflower. Distinction between bacteria and nanoflower difficult. Are there multiple cells attached to the nanoflower? Figure with false colors showing distinct bacteria and nanoflower parts might be useful.
References:
[1]D. C. Dominguez, Molecular microbiology 2004, 54, 291-297.
Author Response
Thank you very much for your review on our manuscript. Your comments and constructive suggestions are very important for us to greatly improve our manuscript and have been very carefully considered point by point. Our responses to your comments can be found in the attachment.

Reviewer 2 Report
1. The most important is the answer a question: what is the correlation between presented method of detection of bacteria in correlation to outer membrane structure of bacteria?
2. Is it possible to use this method for other Enterobacteriaceae detections?
3. The nomenclature of Salmonella should be correct in the whole manuscript text, descriptions of the figures. (Major Compulsory Revisions)
See: WHO Collaborating Centre for Reference and Research on Salmonella. Antigenic formulae of the Salmonella serovars. 2007, 9th edition ( Patrick A.D. Gromont & F.X. Weill)
Therefore, the following examples are correct:
S. enterica subsp. enterica serovar Typhimurium, or
S. enterica serovar Typhimurium or
S. ser. Typhimurium or
S. Typhimurium
When we write: “Salmonella” we have to write : Salmonella sp. or Salmonella spp.
4. How many times experiments were repeated?
5. The term : “concentration of Salmonella” is not appropriate. CFU/mL it is mean: colony forming units of bacteria in mL. The term “ concentration” can be use when we calculate the cells, not in the context of the name of the genus of bacteria
6. In the manuscript I cannot find the section : Discussion, it should be added
Author Response

(The authors gave the same response as above.)

Reviewer 3 Report
The authors have presented in the submitted manuscript an interesting approach to detection of pathogenic bacteria using smart nanocomposite material in combination with simple and robust detection of released ions. The presented method is based on nanocomposite which preparation was published previously and authors significantly improved the final step of the detection procedure using well known and good established electrochemical method – potentiometry with Ca-ISE electrode. Presented results show good linearity of calibration curve for target bacteria in a wide range of concentration with minimal interferention caused by presence of some other bacterial strains. Due to simplicity and reliability the presented method could be interesting for possible application in microbiology laboratories also due to significant shortening of time analysis in comparison with conventional methods. The text is written in a good way, experimental procedures were described with all needed details, and presented results are clearly discussed. I have only one little recommendation – readability of the text could be improved by the explanation of the used abbreviation s in experimental part of the manuscript regardless of their explanation in introduction part of the text (many readers are not reading this part of the articles).
Therefore, I can recommend the submitted manuscript for publication in the Nanomaterials journal.
Author Response
Thank you very much for your review on our manuscript. Your comments and constructive suggestions are very important for us to greatly improve our manuscript and have been very carefully considered. Our responses to your comments can be found in the attachment.

Round 2
Reviewer 1 Report
The authors are able to justify the use of the chemicals Tween 20 and HCl and their potential effects on the cells. They mention that theoretically, the effect of the calcium ions on the response of calcium detection using the lab-on chip method.
But we are still missing confirmation of viabilities during the detection of Salmonella using nanoflowers.
Assays such as adenosine triphosphate quantification, colony forming units (CFU counts), or any other method that helps confirm the integrity of the bacterial membrane should be performed. Regarding the comments on section 2.5: The authors response is “What’s more, most immunoassay could not distinguish between the live and dead one”. The detection might not be hampered, which is true in the case of this publication. However, wether this method affects the membrane integrity during nanoflower-conjugations for example, has still not been proven.
Three nanoflowers were observed for the calculation of the average size of the nanoflowers. Were the three nanoflowers from three different batches, which were synthesized, or are they all from the same batch? A comment on this might be useful to have in the methods section. The answer to this question might help the readers understand the reproducibility of the method of preparation of nanoflowers.
The surface charge data (zeta potential” measurements) should be included in the manuscript to address the issue of unspecific adsorption of these nanoflowers and their respective components with the different elements of the carcasses. Unspecific attachment was expected based on the fact that it is not a pure bacterial culture. The authors should be given more time to include the surface charge data into the manuscript and address all remarks adequatly.
Section3.5: Specificity of the proposed assay:
Use alternative terms such as “minor” or “negligible” instead of “little” when specifying the cross reaction between MAbs/PAbs with E.coli and/or Listeria monocytogenes.
Conclusion: Line 448 in the revised manuscript, has “typhimurium” Instead of “Typhimurium”, starting with a capital T. Please make sure the nomenclature of the bacteria including the italics of the species or the strain name, is even throughout. Also about the bacterial species you specifically used in this publication, please mention the full name of the bacteria in section 2.4 of the manuscript, Salmonella enterica serovarTyphimurium for example.
Author Response
Thank you very much again for your further comments and constructive suggestions on our manuscript. Your comments have been carefully considered point by point and our responses have been attached. Due to the requirement of the journal for us to resubmit our responses within 24 hours, we did some quick tests to answer your questions. More details could be found in the attached file.

Reviewer 2 Report
The nomenclature of Salmonella still need the correction.
Please see that serovar Typhimurium sholud be wirtten without italic ( in the text and fig. 4 and 5)
See: WHO Collaborating Centre for Reference and Research on Salmonella. Antigenic formulae of the Salmonella serovars. 2007, 9th edition ( Patrick A.D. Gromont & F.X. Weill)
The following examples are correct:
Salmonella enterica subsp. enterica serovar Typhimurium, or
S. enterica serovar Typhimurium or
Salmonella ser. Typhimurium or
S. Typhimurium
Author Response
Thank you very much again for your further comment and suggestion on our manuscript. We have corrected the nomenclature of Salmonella in our revised manuscript.
